# Acquisition of the Epistemic Discourse Marker *Wo Juede* by Native Taiwan Mandarin Speakers

Chun-Yin Doris Chen [1,*] , Chung-Yu Wu [1] and Hongyin Tao [2]

1 Department of English, National Taiwan Normal University, Taipei 106308, Taiwan
2 Department of Asian Languages and Cultures, University of California, Los Angeles, CA 90095-1540, USA
* Correspondence: chunyin@gapps.ntnu.edu.tw

**Abstract:** This study examines the use of a fixed expression, *wo juede* (*WJ*) 'I feel, I think', in Taiwan Mandarin in the context of two types of oral production tasks: argumentative and negotiative discourses. The participants consisted of two groups used for comparison: one group of children from Grades 2, 4, and 6, and one group of adults (college students). The results show that both groups were more inclined to utilize *WJ* in argumentative genres than in negotiative genres. Of the seven pragmatic functions associated with *WJ*, the participants all had a strong preference to use *WJ* for the commenting/reasoning function. Developmental patterns gleaned from the data indicate that children's language expands as their age increases. The implications of the findings for cross-linguistic comparison in the realm of epistemic modality are explored in this paper. This study contributes to the study of Chinese morphology by drawing more attention to the acquisition and development patterns of fixed expressions in larger chunks.

**Keywords:** wo juede; fixed expressions; epistemic modality; Taiwan Mandarin; Piaget

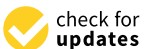



## 1. Introduction

An underexplored lexical and morphological issue in the field of language acquisition and cognitive development is that of fixed expressions, a type of formulaic language. Units larger than the single word can be entrenched as prefabricated expressions with different degrees of lexicalization and specialized functions (Pawley and Snyder 1983; Erman and Warren 2000; Wray 2002; Tao 2020, *inter alia*). With these expressions, a speaker can indicate their positioning with regard to what they are saying, the strength of their commitment to the proposition expressed, and how they view each other's position (Du Bois 2007). *Wo juede* 'I think, I feel' (*WJ*) is such an expression in spoken Mandarin and consists of the first-person singular pronoun *wo* and a mental state verb *juede* lit. 'feel, think'. As a high frequency formula in spoken Chinese (Tao 2005), it has received considerable attention from researchers. This attention is perhaps because it can be used to express both an affective stance (as illustrated in (1)) and an epistemic stance (as illustrated in (2)):

(1) Affective use

| *Wo* | *juede* | *ting* | *gaoxing* | *de.* |
|------|---------|--------|-----------|-------|
| 1SG | feel | quite | happy | PRT |

'I feel quite happy.'                                                    (Lim 2011, p. 269)

(2) Epistemic use

| *Wo juede* | *tamen* | *ye* | *mei* | *you banfa.* |
|------------|---------|------|-------|--------------|
| 1SG think | 3PL | also | NEG | have way |

'I don't think they are able to do anything about it either.' (Self collection)

Example (1) indicates the speaker's internal emotional state (happiness) and is thus an affective use, while Example (2), an epistemic use, indexes a personal opinion based on the speaker's knowledge of the situation.

From a semantic point of view, the verbal component in the formula *juede* has two literal senses: (1) to have a certain feeling; and (2) to have a certain opinion. Together with *wo*, *juede* does not solely index a speaker's personal feelings but extends to the domain of hedged opinions and even textual organization. In terms of modality, even though *wo juede* can be utilized to adopt either an affective or epistemic stance, contextual information (such as the expression of an emotional state such as *gaoxing* 'happy' in (1)), can give important clues as to which stance or modality is being expressed. Note that at least one study has shown that the epistemic stance makes up over 90% of *WJ* use in both Chinese L1 and L2 datasets (Xiao-Desai 2021, p. 687). This matches our observations and has motivated us to focus on epistemic uses in this study. Note also that a versatile expression such as *WJ*, especially in its later development, can be taken to be both an epistemic marker and a discourse cohesive device (Wang 2017).

Research on *WJ* as a fixed expression has been conducted in multiple areas. First, it has shown that the expression has undergone a process of grammaticalization, whereby the clausal structure behaves more like a morphologically single item (Huang 2003; Fang 2005). This echoes the work of Thompson and Mulac (1991) on the English 'I think', which is deemed an 'epistemic parenthetical' due to its bundled nature, syntactic flexibility, and pragmatic extension. Second, from a conversation analytical point of view, Lim (2011) focuses on conversational organization issues related to *WJ*. He notes that when *WJ* is used in connection with an assessment, it is predominantly preposed (86.4%), i.e., placed before a clause or in a conversation sequence-initial position. He identifies two primary conversational organization functions for this patterning, namely, to anticipate possible objections or to work as a joint-assessment initiator. It has been suggested that the less frequent postposed *WJ* functions to express a speaker's assessment with the objective of interactively pursuing the recipients' uptakes (Ford et al. 2002). Along a similar line, Endo (2013) works within the framework of interactional linguistics (Ochs et al. 1996; Selting and Couper-Kuhlen 2001; Ford et al. 2002; Hakulinen and Selting 2005), which concentrates more on actions taken by participants rather than the meanings of linguistic expressions. With 1163 instances retrieved from two everyday conversation corpora, she found that most native speakers prefer utilizing turn-medial (52.5%) and turn-initial (36.8%) positions in multi-unit turns. Her account of this patterning is essentially that at the turn-initial position, *WJ* projects that the interlocutors have potentially conflicting views and foreshadows probable disagreements. In *WJ* prefaced statements, according to Endo, the speaker's dissent becomes less incompatible. Extending this line of research, Wang and Tao (2019) deal with extended conversation sequential functions of *WJ*, noting that it can be used to open a new conversation topic, often initiating a new conversation sequence (Schegloff 2007). Turning to acquisition research, the only investigation so far seems that of Xiao-Desai (2021), which, based on corpora and lexical bundles of *WJ*, compares the stance-taking function of *WJ* in writing by Chinese heritage learners, second language (L2) learners, and native speakers (L1 writers). Among her major findings, L1 users were found to often use *WJ* to express an interpersonal attitude in the (inter)subjective domain, whereas heritage learners were found to use *WJ* primarily for contrastive and causal sequences in the textual domain.

Although the literature reviewed above offers substantial insight into the pragmatics of *WJ*, Chang (2016) plays an important role in the current study. Based on a large corpus, the National Chengchi University Corpus of Spoken Taiwan Mandarin, Chang (2016) develops a taxonomy of the use of *WJ* on the basis of both turn-taking patterns and pragmatic functions. While the next section details identification of the major types of *WJ*, briefly speaking, her major findings include: (1) seven major corpus-driven categories found in the data; (2) that the majority of instances mitigate thoughts in the form of commenting (42%); (3) that the second major category, expressing disagreement, makes up 25% of instances; (4) *WJ* can be differentiated based on pre- and post- positions; and (5) *WJ* is generally used as a mitigating device for a range of interactional tasks. As discussed later, the current study adopts Chang (2016)'s taxonomy as the basis for coding our data. This decision was

made primarily because (1) it is the only study available that has an extensive subcategory system detailed enough to capture the diverse types of use found in conversational data; and (2) it is based on spontaneous Taiwan Mandarin conversation, a variety of Mandarin which has yet to be used as the object of a large-scale study in this area and which also matches the linguistic characteristics of our data.

Regarding the acquisition of the lexicon in Chinese, most studies have focused on specific types such as classifiers (e.g., Chien et al. 2003; Erbaugh 1986; Huang and Chen 2009) and resultative verbs (e.g., Deng 2019). Another area of concern is metaphorical expressions (e.g., Hsu and Chen 2016).

To our knowledge, a fixed expression such as *WJ* has not yet been examined in L1 acquisition of Taiwan Mandarin; thus, we chose to examine and compare the use of *WJ* in three groups of children (Grades 2, 4, and 6) and one group of college students. This study, focusing on child language development, was also motivated by studies of the acquisition pathways of cognitive verbs such as *think* and *know* as well as modal auxiliaries, modal verbs, and modal adverbs such as *can*, *must*, and *probably* in English and other languages. This will be discussed in more detail in Section 4.

Thus, the primary goals were to investigate the frequency and functions of *WJ* as used by native speakers during childhood and shed light on what developmental patterns can be observed in Taiwan Mandarin speaking children. This was achieved by first differentiating two types of task (or genre), argumentative and negotiative, and exploring how genre correlates with the use of this token. We then examined the functional distributions of *WJ* on the basis of Chang (2016). Finally, in this paper we discuss the conversation position tendencies of *WJ* in connected speech.

We hope that the new data and perspectives of this L1 production study contribute to a deeper understanding of this robust discourse token in Mandarin Chinese. Given the nature of the discourse token as a formulaic chunk, we also hope that this study will raise awareness about morphological units that are larger than single words. Such an expansive view of Chinese morphological units will help better our understanding of the wider spectrum of language learning in the context of Chinese.

## 2. Research Design

### 2.1. Participants

As a starting point, elementary school second, fourth, and sixth graders from the Taipei area were chosen as participants for this study. This choice was based on Piaget's (1926, 1952, 1957) cognitive development model, where the concrete-operational stage (7–11 years old) is featured. This seems an ideal developmental stage to investigate the use of linguistic forms that serve to index a speaker's epistemic stance. Comparative studies with children at later developmental stages, such as the fourth stage of propositional or formal operations (Piaget 1926, 1952, 1957), may be conducted in future studies. A total of 45 child participants aged seven to eleven, none with developmental disabilities, were divided into three experimental groups and partook in this research. For a control group, we recruited 15 college students (adults) above the age of 18 who performed comparable tasks. The composition of the two groups is detailed in Table 1.

**Table 1.** Overview of participants.

| Group | | Mean Age | Number |
|---|---|---|---|
| Child groups ('the concrete-operational stage') | G1 (Grade 2) | 7.09 | 15 |
| | G2 (Grade 4) | 9.31 | 15 |
| | G3 (Grade 6) | 11.56 | 15 |
| Adult group | G4 | 24.42 | 15 |

The adults, all native speakers of Taiwan Mandarin, were undergraduates and postgraduates from a college in Taipei.

### 2.2. Methods and Materials

The participants were instructed to engage in group discussions and verbal production tasks. They were led by a researcher in live discussions and debates about a number of topics grouped into two types: argumentative and negotiative, with subtopics shown in Table 2.

**Table 2.** Structure of the production task.

| Mission | Genre | Scenario | Characteristics |
|---|---|---|---|
| M1 | Argumentative | Smart phone | School rules or regulations (formal) |
| M2 | | Casual wear | |
| M3 | Negotiative | Garden party | School events or activities (informal) |
| M4 | | Graduation trip | |

The argumentative topics concerned the subjects' attitudes toward school rules and regulations, where room for negotiation is rather limited. The negotiative topics, on the other hand, involved the subjects planning school events and activities, which left more space for negotiation. To stimulate an animated and reasoned discussion, as well as to keep the children's attention, each topic had been experimentally pre-tested. In addition, a series of prompt questions were also provided to encourage participants to exchange ideas and state opinions. An exemplar example is given in Table 3.

**Table 3.** Example of the production task.

| Prompt heard by the participants: |
|---|
| *Laoshi shuo: Women zhe xueqi yao juban biyeluxing, zonggong you xiamian liang ge didian keyi xuan. Nimen san wei guihua yi xia xiang qu nali, wu fenzhong de xiaozu taolun, huxiang fenxiang geren xiangfa, shijian dao le zai gaosu wo. Yubei qi!* |
| Instructor: We're going to organize a graduation trip this semester, and there are two options from which you can select. In the next five minutes, kindly share your thoughts with your teammates and arrange a trip to one of the two places. Please tell me what you three have decided when the time is up! |
| Picture shown to the participants: |

(A) *Dongwuyuan* Zoo

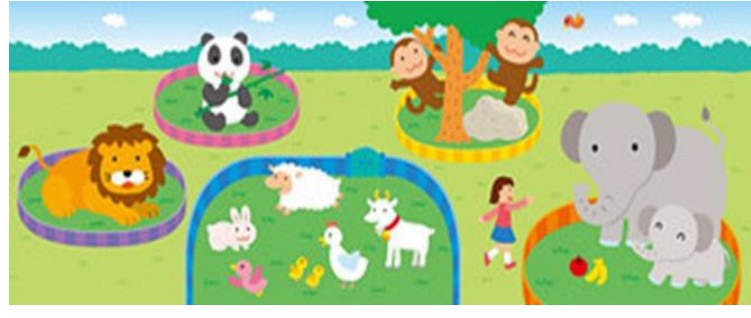

(B) *Shuizuguan* Aquarium

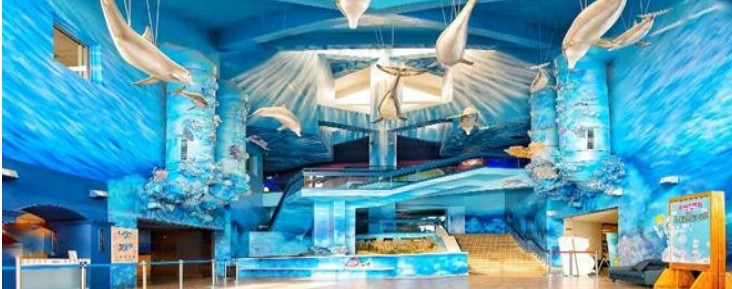

Supplements: Who? Why? When? What? Where? How?

Relatable themes such as campus life seemed to work well for both the young children and college students.

*2.3. Procedures*

Following the recruitment process, three participants of an identical age range were selected for participation at random. The data collection were completed over numerous visits spanning a duration of two weeks. For each experimental task, a semi-constructed dialogue was read to the participants. Each task was completed within five minutes. All four tasks were completed in one session interspersed with three one-minute breaks, thus adding up to a total of approximately half an hour. The participants' natural speech was audio-recorded on a mobile device and was transcribed for annotation. As outlined in Section 2.4, the coding of *WJ* functions followed the system used in Chang (2016). To improve inter-rater reliability, two raters were invited to code all instances of *WJ*. When there were disagreements, a third rater was invited to look at the data so that a consensus could be reached.

*2.4. Functional Types and Identification Criteria*

Chang (2016) proposes the seven functional categories of *WJ* as follows: T1 (agreeing), T2 (disagreeing), T3 (commenting), T4 (speculating), T5 (suggesting), T6 (concluding), and T7 (expressing afterthoughts). (Note that T1 and T2 can be collapsed into one, yielding a total of six, if polarity is to be ignored.) Examples from our corpus are used below to illustrate these categories. We wish to emphasize that it can be difficult to put instances of *WJ* into different functional categories just by themselves. These categories exist mainly by examining the collocating tokens and/or the discourse positions that they adapt, which are summarized in Table 4.

**Table 4.** Functional categories and their correlating features in discourse.

| Type | Function | Formula |
|------|----------|---------|
| T1 | Agreeing | [an agreeing particle (e.g., *duiya/en'en*)] + [*wo juede*] + ([*ye*]) + [agreement] |
| T2 | Disagreeing | [a transition word (e.g., *danshi/keshi*)] + [*wo juede*] + [disagreement] |
| T3 | Commenting/Reasoning | ([*yinwei*]) + [*wo juede*] + [the topic on which they are commenting] + [comment] |
| T4 | Speculating | [*wo juede*] + [the topic on which they are speculating] + [*yinggai/keneng*] + [speculation] |
| T5 | Suggesting | [*wo juede*] + [the topic about which they are making suggestions] + [suggestion] |
| T6 | Concluding | ([*suoyi*]) + [*wo juede*] + [conclusion] |
| T7 | Expressing Afterthoughts | [statements] + [clause-final *wo juede*] |

T1: Agreeing

Agreement tokens were often used to express the speaker's congruent stance with what was being expressed in the prompt by the researcher.

(3) G3-S10: *Yinggai bu yong jinzhi ba, yinwei youshihou shangke laoshi yehui yongdao a.*
perhaps NEG need ban PRT because sometimes in class teacher also use PRT
'Perhaps (smartphones) need not be banned at schools because teachers sometimes also use them (for teaching purposes) in class.'

G3-S11: *Duiya, wo juede ye bu xuyao jinzhi. Bu yong de shihou zai ba ta shouqilai jiu hao.*
exactly 1SG think also NEG need ban NEG use GEN time then BA 3SG put away so be it
'Exactly, I think smartphones should be allowed in the classroom. We can put them aside whenever they are not needed.'

(M1_G3_Team D)

As exemplified by the above extract, in agreeing sequences *WJ* was often preceded by an agreeing participle, such as *dui ya* 'yes, right + PRT', and followed by an elaboration. In most cases, agreeing tokens of *WJ* occurred in the clause-initial position.

T2: Disagreeing

By contrast, in disagreeing tokens, *WJ* was usually preceded by a contrastive or transition conjunction such as *danshi* in (5) and *keshi*, followed by the speaker's disagreement.

Similar to the agreeing, disagreeing tokens also tended to occur in the clause-initial position, as shown in (4).

(4) G2-S13: *Wo xiang qu shuizuguan. Nali you henduo wo mei kan guo de dongwu.*
1SG want go aquarium there have many 1SG NEG see PRF GEN animal
'I want to go to the aquarium. There are many (marine) animals there that I have never seen before.'

G2-S14: *Danshi <u>wo</u> <u>juede</u> biye luxing yao you haowande.*
but 1SG think graduation trip need have fun
'But I think graduation trips are supposed to go to places with lots of fun things.'

(M4_G2_Team E)

### T3: Commenting/Reasoning

Commenting is closely related to reasoning, where the speaker gives a rationale for proposing certain ideas. In such cases, conjunctions such as *yinwei* 'because' occasionally preceded *WJ*, and also tended to be used in the clause-initial position, as illustrated below.

(5) E: *Weishenme ni xiangyao meitian chuan bianfu?*
why 2SG want every day wear casual clothes
'Why do you want to wear casual clothes every day?'

G1-S7: *Yinwei <u>wo</u> <u>juede</u> chuan bianfu bijiao hao, bu yong yizhi xiang mingtian yao*
because 1SG think wear causal clothes COMP good NEG need all the time think tomorrow want
*chuan sheme.*
wear anything
'Because I think it's better to wear causal clothes. I do not need to worry about what I am going to wear the next day.'

(M2_G1_Team C)

### T4: Speculating

Speculations indicate a speaker's uncertainty about a proposed state of affairs. Uncertainty was often expressed with modal adverbs such as *keneng* 'perhaps, likely' and *yinggai* 'should, likely'.

(6) G2-S7: *Mai shiwu dehua lengdiao jiu bu haochi le.*
sell food if become cold then NEG delicious PRT
'This food is delicious only when it is hot.'

G2-S8: *<u>Wo</u> <u>juede</u> yinliao keneng henduo ren hui mai.*
1SG think drink might many people will buy
'I think many people might be interested in buying drinks.'

(M3_G2_Team C)

Additionally, in these cases, the clause-initial tokens of *WJ* dominated.

### T5: Suggesting

These cases are straightforward with *WJ* simply followed by what the speaker suggests.

(7) G3-S1: *Jiushi dai qu dehua cha ziliao hen fangbian.*
that is bring go if look for information very convenient
'If you bring your smartphone with you, it will be very convenient for looking up information.'

G3-S2: *<u>Wo</u> <u>juede</u> laoshi keyi zhiding mouge shijian rang ni qu shiyong, ranhou wenti ziji shangke*
1SG think teacher can impose certain time let 2SG go use later question self in class
*xian jilu xialai dao shihou zai cha.*
before note down until then again look for
'I think teachers can fix a time to use smartphones so that you may find answers to questions you jotted down in class.'

(M1_G3_Team A)

Once again, *WJ* in clause-initial positions rather than final positions prevailed in both groups.

T6: Concluding

The concluding function draws an extended discussion to an end by giving the gist of a proposal. This is illustrated in (8).

(8) Adult
group-S4: *Buguan*         *tianqi*   *leng*   *bu*   *leng*   *re*   *bu*   *re,*   *ni*   *zai waimian guang*   *le*   *hen*   *jiu,*
    regardless of   weather   cold   NEG   cold   hot   NEG   hot   2SG   in   outside   wander   PRT   very   long
    *yiding hui kouke*     *xiangyao*       *he*           *dongxi.*   *Suoyi,*   <u>*wo*</u>   <u>*juede*</u>   *leng*   *yin*   *bijiao*   *neng*   *manzu*
    must will thirsty      want      drink   something   hence   1SG   think   cold   drink   COMP   can   satisfy
    *dazhong xuqiu.*
    public   need
'Regardless of the weather, you are more likely than not to feel thirsty if you wander off somewhere for a long period of time. Hence, I think (selling) cold drinks would better satisfy the public's needs.'

(M3_Adult group_Team B)

T7: Afterthoughts

The last category in Chang (2016) and used here is 'afterthoughts'. Although the term 'afterthoughts' has been disputed due to its negative connotations (for recent studies from a conversation analytic standpoint, see, e.g., Lim 2011; Luke 2012), we kept it here following Chang (2016). In such cases, *WJ* is used after a clause is completed as shown in (9).

(9) G3-S1: *Dai*     *shouji*       *juisuan*     *meiyou yaocha*    *ziliao,*       *shangxue*     *fanxue*
    bring   smartphone   even though    NEG   look for   information   go to school   after school
    *dadianhua*          *ye hao,*      *ruguo*    *you*   *shenmeshi*   *yao*   *lianluo,*   <u>*wo*</u>   <u>*juede*</u>*.*
    make a phone call   also good      if      have   something   need   contact   1SG   think
'Even though you may not use your phone to look up information, being able to make phone calls before or after school if needed is also good, I think.'

(M1_G3_Team A)

Table 4 summarizes the key features of the seven functional categories.

Overall, 245 instances of *WJ* were identified from the production samples provided by the four age groups, the three child experimental groups (G1~G3) and the one adult control group.

### 3. Results and Discussion

The discussion below deals with the genre effect, functional distribution, and conversation position tendencies.

#### 3.1. Genre Effect

As mentioned earlier, we divided the genres into argumentative and negotiative types. It was interesting to find that *WJ* was more frequently employed in argumentative than in negotiative genres, as shown in Figure 1. As we can see, there are about seven times as many instances of *WJ* in argumentative genres than in negotiative genres in G1 (7 vs. 1), twice as many in G2 (20 vs. 10), and almost six times as many in G3 (29 vs. 5). Likewise, the results of the chi-squared test show a significant within-group difference in *WJ* between the two aforementioned genres (G1, $p = 6.382 \times 10^{-14} < 0.05$; G2, $p = 0.0008561 < 0.05$; G3, $p = 1.689 \times 10^{-12} < 0.05$). This in turn suggests an uneven distribution of *WJ* across the two genres.

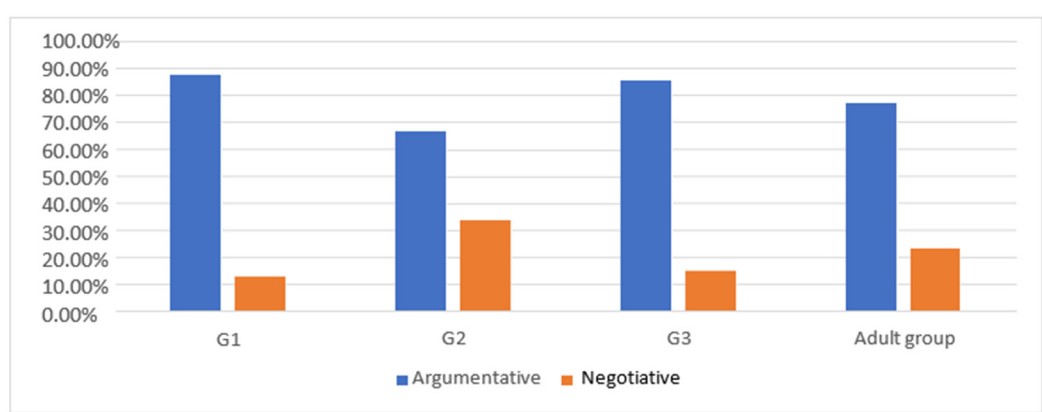

**Figure 1.** Distribution of *WJ* between argumentative and negotiative genres by the participants.

A number of characteristics of argumentative discourse may be taken to have contributed to the much higher frequency of *WJ* in this genre. There is no doubt that argumentative discourse often involves putting forward arguments or counterarguments in order to support one's stance (e.g., Ellis 2015). Regardless of a participants' positive or negative stance, it was usually a matter of concern over which position prevailed. In other words, the argumentative genre leans toward stronger interpersonal involvement (Xiao-Desai 2021) than the negotiative genre as the latter can focus on personal (subjective) opinions without necessarily or explicitly negating or supporting the other party's stance. As a lexicalized marker, *WJ* can help make a personal stance explicit, which is necessary for the interpersonal negotiation of stances in argumentative discourse.

However, no significant between-group (between the child and adult groups) difference was found ($p > 0.05$) in either genre. Despite this, what stands out in Table 5 is that, in comparing the child groups, G2 was significantly different from G1 (G2 vs. G1, $p = 0.002092 < 0.05$) as well as G3 (G2 vs. G3, $p = 0.007222 < 0.05$). This result was rather unexpected.

**Table 5.** Distribution of *WJ* in argumentative and negotiative genres.

| Group | G1 | | G2 | | G3 | | Adult | |
|---|---|---|---|---|---|---|---|---|
| Genre | *N* | % | *N* | % | *N* | % | *N* | % |
| Argumentative | 7 | 87.50% | 20 | 66.67% | 29 | 85.29% | 133 | 76.88% |
| Negotiative | 1 | 12.50% | 10 | 33.33% | 5 | 14.71% | 40 | 23.12% |
| Total | 8 | 100% | 30 | 100% | 34 | 100% | 173 | 100% |

A possible explanation involves considering one of the fourth graders in G2. This primary school pupil produced four negotiative instances of *WJ*, accounting for almost half of the total number produced by the group, and therefore, as is well exemplified in the following excerpt, skewed the results. Most students in the G2 group tended to deliver short replies, while this more gifted student behaved remarkably in producing more sophisticated answers.

(10) G2-S8: *Ni   weishenme xiang mai  re    de     dongxi?*
      2SG    why    want sell hot GEN  something
      'Why do you want to sell hot food?'

G2-S7: *Yinwei <u>wo</u> <u>juede</u> ruguo zai  bu    leng  bu    re   de    tianqi,    ni    shenti   de       wendu*
      because 1SG think  if    in NEG  cold NEG hot GEN  weather 2SG   body   GEN temperature
      *yeshi gangganghao, ruguo  ni   chixiaqu  bing  de      dongxi dehua, ni  jiuhui rongyi   ganmao*
      also     normal      if   2SG consume cold GEN something  if   2SG   will   easily  catch a cold
      *ya, huozheshi bu     shufu     zhileide.*
      PRT    or     NEG comfortable et cetera
      'I think it is because of such ideal weather that you are at the average normal body temperature. If you consume something cold right away, you will easily catch a cold or feel unwell.'

(M3_G2_Team C)[1]

Given this anomalous case, we removed the four tokens from the negotiative genres and repeated the statistical test. Table 6 illustrates a modified version of Table 7 in which a non-significant between-group difference in argumentative and negotiative genres appears.

**Table 6.** Revised between-group differences of genre effect among the participants.

| Comparison | Argumentative | | Negotiative | |
|---|---|---|---|---|
| | $x$ | $p$ | $x$ | $p$ |
| G1 vs. G2 | 0.6808 | 0.4093 | 3.146 | 0.07611 |
| G1 vs. G3 | 0.028266 | 0.8665 | 0.1795 | 0.6718 |
| G1 vs. Adult group | 0.68612 | 0.4075 | 3.1663 | 0.07517 |
| G2 vs. G3 | 0.43189 | 0.5111 | 1.8538 | 0.1733 |
| G2 vs. Adult group | $1.0403 \times 10^{-5}$ | 0.9974 | $3.4632 \times 10^{-5}$ | 0.9953 |
| G3 vs. Adult group | 0.43614 | 0.509 | 1.8696 | 0.1715 |

**Table 7.** Between-group differences of genre effect among the participants.

| Comparison | Argumentative | | Negotiative | |
|---|---|---|---|---|
| | $x$ | $p$ | $x$ | $p$ |
| G1 vs. G2 | 2.8144 | 0.09342 | 9.4674 | 0.002092 * |
| G1 vs. G3 | 0.028266 | 0.8665 | 0.1795 | 0.6718 |
| G1 vs. Adult group | 0.68612 | 0.4075 | 3.1663 | 0.07517 |
| G2 vs. G3 | 2.2816 | 0.1309 | 7.217 | 0.007222 * |
| G2 vs. Adult group | 0.72619 | 0.3941 | 1.8467 | 0.1742 |
| G3 vs. Adult group | 0.43614 | 0.509 | 1.8696 | 0.1715 |

* A significance level of $p < 0.05$.

In particular, the two statistically significant between-group comparisons shown in Table 7 turned out to be statistically non-significant (G2 vs. G1, $p = 0.07611 > 0.05$; G2 vs. G3, $p = 0.1733 > 0.05$). That is, the three child groups exhibited comparable patterns of usage to the adult group regarding the frequency of *WJ* in *both* argumentative and negotiative use. Yet, qualitatively speaking, a few individual differences such as choice of words, as elucidated above in (10), were found in each child group.

With reference to within-group and between-group comparisons of genres, the child groups were found to be quite adult-like, preferring argumentative to negotiative discourse in the use of *WJ*. From the perspective of acquisition, our findings also demonstrate that argumentative discourse was easier for the child participants. This is probably because the contextual prompts primed the subjects to express their opinions in response to the other party's stance as explicitly as possible, a requirement which tokens of *WJ* can help accomplish. This supports the findings of Xiao-Desai (2021) based on the L1 speaker writing samples.

### 3.2. Functions of WJ

Table 8 schematically displays the distribution patterns of the seven functions of *WJ* across the child and adult groups. It was striking that T3 (commenting) once again made up the largest proportions (G1, 75.00%; G2, 63.33%; G3, 70.59%; the adult group, 56.07%). Additionally, an upward growth trend was found as the age of the participants increased. Namely, the older children employed a wider range of *WJ* functions. For instance, G1 employed only three *WJ* functions (i.e., T1~T3), G2 employed five functions of *WJ* (T1~T5), and G3 used six *WJ* functions, with only T7 not found in their utterances. These observations seem to show that the child groups move toward a functionally diverse pattern as they age, with G1 being the least developed in terms of diversified usage patterns, followed by G2 and G3. Naturally, the adult group employed the most diversified usage pattern. It is also worth noting that T6 (concluding) was the key category that differentiated the three child groups from the adult group as G1, G2, and G3 presented no usages of concluding (T6).

**Table 8.** Distributions of the functions of *Wo Juede* among the participant groups[2].

| Type | G1 | | G2 | | G3 | | Adult Group | |
|---|---|---|---|---|---|---|---|---|
| | *N* | % | *N* | % | *N* | % | *N* | % |
| T1 | 1 | 12.50% | 3 | 10.00% | 1 | 2.94% | 10 | 5.78% |
| T2 | 1 | 12.50% | 3 | 10.00% | 2 | 5.88% | 14 | 8.09% |
| T3 | 6 | 75.00% | 19 | 63.33% | 24 | 70.59% | 97 | 56.07% |
| T4 | 0 | 0.00% | 2 | 6.67% | 3 | 8.82% | 18 | 10.40% |
| T5 | 0 | 0.00% | 3 | 10.00% | 3 | 8.82% | 15 | 8.67% |
| T6 | 0 | 0.00% | 0 | 0.00% | 0 | 0.00% | 17 | 9.83% |
| T7 | 0 | 0.00% | 0 | 0.00% | 1 | 2.94% | 2 | 1.16% |

In general, as far as the functions of *WJ* are concerned, T3 (commenting) exhibits the most prominent pattern across all groups. This is congruent with Chang (2016), who used natural conversations from the NCCU Corpus of Spoken Mandarin as her data source. To a certain degree, the present study and Chang's complement each other in the sense that her data are overwhelmingly negotiative in nature, while ours supplemented this with the argumentative genre.

Table 9 ranks the seven functions in decreasing order. A chi-square test was used to scrutinize the participants' within-group differences in function. The results provide a reliable indication of how T3 significantly differs from many others, as can be seen in the following: within G1, T3 vs. T1/T2, $p = 2.365 \times 10^{-11} < 0.05$; T3 vs. T4/T5/T6/T7, $p = 2.2 \times 10^{-16} < 0.05$; within G2, T3 vs. T1/T2/T5, $p = 4.732 \times 10^{-10} < 0.05$; T3 vs. T4, $p = 1.269 \times 10^{-11} < 0.05$; T3 vs. T6/T7, $p = 1.748 \times 10^{-15} < 0.05$; within G3, T3 vs. T1/T7, $p = 3.04 \times 10^{-15} < 0.05$; T3 vs. T2, $p = 1.363 \times 10^{-13} < 0.05$; T3 vs. T4/T5, $p = 4.158 \times 10^{-12} < 0.05$; T3 vs. T6, $p = 2.2 \times 10^{-16} < 0.05$.

**Table 9.** Patterns of *wo juede* functions among the participants[3].

| G1 | T3 > T1 = T2 > T4 = T5 = T6 = T7 |
|---|---|
| G2 | T3 > T1 = T2 = T5 > T4 > T6 = T7 |
| G3 | T3 > T4 = T5 > T2 > T1 = T7 > T6 |
| Adult group | T3 > T4 > T6 > T5 > T2 > T1 > T7 |

Statistical evidence brought to light apparent discrepancies in the mathematical symbols 'more than' and 'equal to' for the four patterns. First, there were only three groupings for G1 since T1 vs. T2 as well as T4~T7 showed no apparent discrepancy. Likewise, T1, T2, T4, and T5 were statistically insignificant for G2, indicating that these functions were used with equal frequency. As for G3, it was noted at the outset that no more than two

groupings need be formed to make G3 exactly the same as the adult group, and so T4 vs. T5 ($p = 1 > 0.05$), T5 vs. T2 ($p = 0.4432 > 0.05$), T2 vs. T1 ($p = 0.3222 > 0.05$), T1 vs. T7 ($p = 1 > 0.05$) and T7 vs. T6 ($p = 0.08641 > 0.05$) were statistically non-significant within G3. Yet again, T3 attracted the highest usage frequency in all the groups, creating a separate category of its own. It is no surprise that T3 consistently outperformed all the others, indicating that *WJ* both semantically and pragmatically aligns with the concept of expressing one's comments.

The between-group differences between the child groups and the adult group give a much more balanced picture of children's developmental stages. First, G1 and G2 differed significantly in their T4 frequency of use ($p = 0.009805 < 0.05$) as well as T5 ($p = 0.001565 < 0.05$). This is because neither function appeared prior to G2. Next, though the probability values between G2 and G3 insinuate a significant difference in the use of T1 ($p = 0.04969 < 0.05$), this function was already present in G1. What sharply distinguished G2 from G3 then was T7, present in G3 but not in G2, even though the two groups differed non-significantly in their use of T7 at the $p > 0.05$ level ($p = 0.08641$). Furthermore, G3 and the adult group again differed significantly in their use of T6 ($p = 0.001717 < 0.05$), there being zero instances of concluding *WJ* across the child groups. In brief, the presence and absence of T4, T5, T7, and T6, then, are important to note.

On closer inspection, a more circumspect interpretation of the research findings is in order. First, the complete absence of T4 and T5 sets G1 apart from G2 and may be attributed to cognitive factors as the speculating function of (T4) and the suggesting function (T5) can be seen as more cognitively demanding. According to Piaget's cognitive development model, hypothetical operation does not take place until the fourth stage in children of ages 11–12 to 14–15 (Piaget 1926, 1952, 1957). In addition, a considerable amount of working memory resources may be needed to exchange this kind of information (e.g., Sweller 1988). Overall, the psychology literature suggests that the lack of these two functions in the child groups may be accounted for by a lack of higher-order thinking skills (e.g., Anderson and Krathwohl 2001).

Second, the presence of T7 from G3 onwards may be attributed to a decrease in egocentrism with the passage of time and an increased ability to reason by hypothesis (e.g., Piaget 1926). This is partially related to the defining characteristics of clause-final *WJ* for expressing afterthoughts as a result of the subjects' (inter-)active considerations of recipient design (e.g., Sacks et al. 1974). With respect to T4, T5, and T7, it seems indicative that children become more conscious of differentiating between the self and other by this stage in their development. Compared with the lower and intermediate grades, there was an encouraging sign that the upper grades had started to put themselves in others' positions (hypothetically). Instead of using *WJ* in the turn-initial position, the G3 group and above made the best of turn-final *WJ* to shape something that was not originally integrated but was thought of or added at a later temporal point. This pattern, anomalous in structure and demanding in online processing, can be challenging to the child G1 and G2 groups as, in Lim (2011)'s contention, the post-positioned *WJ* is a rather sophisticated conversation strategy, expressing a speaker's stance/assessment with the objective of interactively pursuing the recipients' uptakes on previously articulated materials.

Notwithstanding the variety of pragmatic functions which increase with age, Piaget's (1952) theory of cognitive development holds that most children at the concrete-operational stage still experience difficulty with hypothetico-deductive reasoning, a process of reasoning from either statements or premises to logical conclusions. The isolated usage, or even the non-occurrence, of T6 (concluding) *WJ* in the child groups is highly likely to be due to the level of cognitive development. Simply put, indirect observations of an inability to think on a larger scale and an ability to think on a smaller scale (more inductive; less deductive) by children aged 7 to 11 closely parallel classical theories of cognitive development (e.g., Piaget 1952). In progressing from inductive to deductive logic, they are growing into more adult-like individuals. In short, in terms of degrees of difficulty, T3 (commenting) had a greater frequency of use, but T6 (concluding) had a lower frequency of use by the child

groups. The former is an early acquisition and the latter is a late acquisition. Nevertheless, an acquisition order of the other functions is cautiously presumed.

*3.3. Wo Juede and Conversation Position Tendencies*

Having examined the genre association and functional distribution patterns in the child and adult groups, we now examine some position tendencies. By position tendencies, we mean the position of *WJ* in the clause or turn constructional units (TCUs). While clause is a standard syntactic category typically involving a subject and predicate, TCU is a broader conversation structural unit which refers to utterances that make up a speaker's turn in talk-in-interaction (Sacks et al. 1974). Here we found that of the seven categories, the concluding and after-thought categories were the only ones that showed differences between the child and adult groups.

Regarding the concluding use, participants in the child groups often drew conclusions with a connective (e.g., *suoyi* 'so') rather than *WJ* to reformulate or link their ideas, while those in the adult group employed *WJ* in clause-initial position. The adult case is shown in Example 11, seen earlier but copied again below.

(11) Adult
group-S4: *Buguan         tianqi    leng   bu   leng  re   bu   re,   ni   zai waimian guang   le    hen   jiu,*
regardless of  weather  cold  NEG  cold  hot  NEG  hot  2SG  in  outside wander PRT  very  long
*yiding hui  kouke  xiangyao      he         dongxi.  Suoyi,  wo   juede leng   yin   bijiao   neng  manzu*
must  will thirsty    want    drink something hence 1SG  think  cold  drink COMP  can    satisfy
*dazhong xuqiu.*
public  need
'Regardless of the weather, you are more likely than not to feel thirsty if you wander off somewhere for a long period of time. Hence, I think (selling) cold drinks would better satisfy the public's needs.'

(M3_Adult group_Team B)

The next category is afterthoughts, where we see that while adults often place *WJ* at the end of the clause or TCU, children rarely do so. This is illustrated in (12).

(12) Adult
group-S7: *Ni   yao  zenme gen   ni    de        shouji      gongtong xiangchu, zhege shi   bu   shi keyi    dajia*
2SG  need how and 2SG  GEN  smartphone  together get along  this is   NEG  is  can everyone
*yiqi    zuo  taolun, danshi  bu   shi yingai  yao  xiao   de  shihou  jiu   jiao,   wo   juede.*
together  do  discuss  but  NEG  is  should  need young GEN  time  then teach 1SG  think
'How to live in perfect harmony with your smartphone is something worth discussing. Yet, we should have taught them such a lesson when they were young, right? I think.'

(M1_Adult group_Team C)

In short, while there was much similarity between the child and adult groups, differences in discourse structural configurations were observed. While grammarians have tended to focus on the structural anomaly of T7, the above examples also show that the use of *WJ* is sensitive to turn-taking functions and utterance-initial/-final positions, as demonstrated in Lim (2011); Endo (2013); and Wang and Tao (2019).

**4. Conclusions**

The results from our Taiwan Mandarin L1 production data show that both the child and adult groups were more inclined to utilize *WJ* in argumentative genres than in negotiative genres. As for the seven pragmatic functions associated with *WJ*, all the participants displayed a strong preference to use *WJ* for T3 (commenting/reasoning). Developmental patterns indicate that from the lower grades (i.e., G1) to the intermediate grades (i.e., G2), two functions (T4, speculating and T5, suggesting) are acquired. Next, from the intermediate grades to the upper grades (i.e., G3), the use of *WJ* to express afterthoughts (T7) is acquired. However, none of the child groups had acquired the concluding *WJ* (T6),



suggesting that children at the concrete-operational stage are still limited in their ability to express hypothetico-deductive logic and are limited instead to inductive logic. Overall, G1 had the least diversified pattern of *WJ* usage whereas the adult group had the most.

This study provides fresh data to feed the debate surrounding child language acquisition of epistemic modality. Most early work on epistemic modality focused on processing modal auxiliaries or modal verbs such as *might* and *will* (Armstrong 2020), *can, cannot, (doesn't) have to* (Gonsalves 1999), and *must* and *could* (Cournane 2021); or modal adverbs (e.g., *maybe, probably*, see Cournane 2021). Crosslinguistic investigations exhibit a similar tendency, as shown in work such as *mo'ci* 'can' and *morati* 'must' in Bosnian/Croatian/Serbian (Veselinovic 2019), or *chce* 'want' in Polish (Smoczynska 1993). Rarely do we see studies on nearly lexicalized epistemic expressions such as *wo juede* (or *I think/I guess* in English).

Our findings confirm some of the general assumptions of the field, that is, there is an 'Epistemic Gap' (Cournane 2021) in child language development, which states that children generally understand the root meaning of modal lexical items earlier than their epistemic extensions. For example, it has been shown that until the age of three, constructions involving cognitive verbs in epistemic constructions such as *(I) think, (I) feel* are non-existent (e.g., Bretherton and Beeghly 1982; Shatz et al. 1983), a situation which changes at around the age of four (see, e.g., Johnson and Maratsos 1977; Moore et al. 1989; Moore and Furrow 1991; Naigles 2000). In terms of modal lexical items, Gonsalves (1999, p. 2) reports delayed comprehension of modal verbs in the epistemic context compared with better comprehension in the deontic and dynamic contexts, suggesting that 'advances in epistemic modal verb meaning await changes in the nonlinguistic domain, such the logical understanding of the possibility/necessity distinction.'

Likewise, this study contributes to the study of Chinese morphology by raising awareness about acquisition and development patterns in the area of fixed expressions/formulaic language. Most existent studies in the Chinese lexicon have focused either on single lexical items or large syntactic structures, and constructional units with fixed expressions have not attracted as much attention as they deserve. Yet research from multiple fields has demonstrated the pervasiveness of prefabricated expressions (Pawley and Snyder 1983; Erman and Warren 2000; Wray 2002; Tao 2020, *inter alia*). By examining *WJ* in L1 production, one of many such tokens frequently used by speakers in spoken conversation, we hope to have shown that fixed expressions are a promising area of investigation.

Concerning the mechanism of child language development in the realm of epistemic modality, our study provides potentially useful insights for cross-linguistic comparisons. Our appeal to the cognitive development stages where hypothetico-deductive logical operation is assumed to come late for some of the subjects substantiates some of the lexical item-based results. For example, Smoczynska's (1993) work on Polish epistemic modality in young Polish children (ages 1:0–3:1) concludes that although epistemic modal usage is at first tied to the immediate situation, it gradually becomes more decontextualized. Our findings concerning a lexicalized epistemic marker agree in general with this statement as hypothetico-deductive logical operation can also be considered as belonging to the category of the more abstract and less immediate.

In terms of practical pedagogical applications, a few suggestions may be made. For example, given that the participants all strongly favored the argumentative *WJ* above the negotiative *WJ*, language instructors may construct certain discourse scenarios for classroom activities with the object of assessing the ability to carry out various *WJ* functions. In addition, *WJ* formulas could be consciously included in teaching and learning materials. Finally, whenever applicable, instructors could develop portfolios to track in both the spoken and written modes developmental sequences of using *WJ* and similar epistemic formulas in diverse contexts.

To conclude, we must also acknowledge that as an initial project this study has a number of limitations. One is the modest sample size. More extensive studies will be needed to uphold any legitimate claims. Second, this study relies on Chang's (2016) functional taxonomy of *WJ*. A more refined classification system may be needed to incorporate up-

to-date findings from conversation analysis and interactional linguistics. Third, regarding acquisition of the concluding type of *WJ*, a potentially profitable area of research may be to focus on the formal-operational stage (Age 11+) by recruiting junior or even senior high school students. Finally, in view of the apparent similarities between *WJ* and *I think*, a promising avenue in search of cross-linguistic transfer could be pursued by recruiting Chinese–English bilingual children.

**Author Contributions:** Conceptualization, C.-Y.D.C. and C.-Y.W.; methodology, C.-Y.D.C., C.-Y.W. and H.T.; software, C.-Y.W.; validation, C.-Y.D.C. and C.-Y.W.; formal analysis, C.-Y.D.C., C.-Y.W. and H.T.; investigation, C.-Y.D.C., C.-Y.W. and H.T.; resources, C.-Y.D.C. and C.-Y.W.; data curation, C.-Y.W.; writing—original draft preparation, C.-Y.D.C. and C.-Y.W.; writing—review and editing, C.-Y.D.C., C.-Y.W. and H.T.; visualization, C.-Y.W.; supervision, C.-Y.D.C.; project administration, C.-Y.D.C.; funding acquisition, C.-Y.D.C. All authors have read and agreed to the published version of the manuscript.

**Funding:** This work was financially supported by the 'Chinese Language and Technology Center' of National Taiwan Normal University (NTNU) within the framework of Higher Education Sprout Project by the Ministry of Education (MOE) in Taiwan. The author Hongyin Tao would also like to acknowledge the support of Fulbright Canada and the University of Alberta for the Research Chair fellowship support during the writing of this paper in Spring 2022.

**Institutional Review Board Statement:** Ethical review and approval were waived for this study as its activities were in the forms of interview and observation of public behavior (including auditory recording), thus posing no greater than minimal risk to the participants.

**Informed Consent Statement:** Informed consent was obtained from the adult participants and the parents of all the child participants involved in the study.

**Data Availability Statement:** The data presented in this study are available on request from the corresponding author.

**Acknowledgments:** The authors wish to thank the following people for their valuable contributions: the children and adult subjects for their participation in the project; Willy Wang, Eliot Huang, Ruby Lu, and Amy Huang for research assistance; the five anonymous referees for their constructive feedback; and Andrew Dodd and Kerry Sluchinski for editorial assistance.

**Conflicts of Interest:** The authors declare no conflict of interest.

## Notes

[1]   Note that there were five teams (Teams A~E) in each of the Child (G1~G3) and Adult (Adult group) groups. The participants of each group were numbered from S1 to S15.

[2]   A note on the statistics used here: While the sample sizes are small and there is only one cell in G1 under 5, the other cells are above 5. In Table 8, though more cells under 5 resulted when the functions of *WJ* were examined, our purpose was to conduct a within-group comparison to investigate the functional variation in each age group, and a between-group comparison to look at how age affected the use of each function type. As such, we believe that the current method remains applicable and effective. The same reason also applies to the first research question, designed to examine the genre effect and age factor. We thank an anonymous reviewer whose comments prompted us to clarify the issues involved here.

[3]   Results from the chi-square test for each one of two given probabilities show rankings with novel groupings of patterns underlined. The underline serves to group together functions whose rank order is either equivalent or statistically insignificant.

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
