# Peer review of "Acquisition of the Epistemic Discourse Marker Wo Juede by Native Taiwan Mandarin Speakers"

_languages, doi:10.3390/languages7040292_

Round 1
Reviewer 1 Report
(Also see the attachment)
Journal: Languages
Review for Acquiring the Epistemic Discourse Marker Wo Juede in L1 Taiwan Mandarin
Date: 20220612
Decision: Reconsider after major revision
Overall comments:
• This work aims to investigate the use of the formulaic/fixed expression wo juede ‘(lit.) I think’ in the context of Taiwan Mandarin by comparing its use of children (Grade 2, 4 and 6) with that of adults from the perspective of language acquisition and cognitive development. The findings are rather thought-provoking and provide implications for our understanding of cognitive development and language use. Given the major comments listed below, I recommend that this manuscript be evaluated as ‘Reconsider after major revision’.
a. The literature review in Section 1 is far from complete in the sense that many crucial issues are not properly acknowledged, as listed below, making it hard to follow the research gap the author attempts to identify.
• What type of epistemic modality does the author intend to follow or what kind of epistemic modality is wo juede associated with? The author seems to assume the encoding of certain epistemic modality but what it is remains unclear.
• Any acquisition study that addresses the acquisition of formulaic/fixed expressions in Mandarin or other languages? Or, how are formulaic/fixed expressions treated in the field of cognitive development?
• The review of the five studies from line 45 to 54 is heavily simplified. What are their findings and implications that point to a research gap the author wishes to fill?
It is suggested that the author should add completeness to the literature review in order to convince the reader of the research gap this work aims to fill. Otherwise, this paper seems to simply re-examine Chang’s (2006) seven functions of wo juede by considering the use of wo juede from the perspective of language acquisition.
b. It is concluded from this study that both the child and the adult groups preferred to use wo juede in argumentative genres rather than in negotiative genres. Obviously, the finding confirms that the use of wo juede is genre-specific or genre-induced. But it is not clear how the finding is interpreted from the perspective of language acquisition? For example, language acquisition of cohesive devices or discourse organization?
c. The discussion in Section 3 is fairly self-contained and shows that the results seem to be Mandarin-specific. It is suggested that the author could consider providing crosslinguistic evidence or discussions from language acquisition or cognitive psychology to demonstrate how the findings in the current work can be viewed from a broad perspective, with an effort to clearly demonstrate how the use of wo juede develops in congruence with cognitive development.
d. It is suggested that the manuscript should be first proofread by the native speaker of English. The English is in need of improvement, as there are typos and parts of the phrasing remains downright incomprehensible at various points
Major and minor comments are given as follows for reference.
Detailed comments:
• Ex. (1): I am wondering why (1) cannot be interpreted as saying that the speaker thinks that s/he is happy based on the speaker’s knowledge of the state. Any diagnostic to exclude this reading?
• As (1) is cited from Lim (2011), it is suggested that the author should mention how the affective use is defined.
• In Section 1, no discussion about epistemic markers is provided. As a reader, I am not able to understand what types of epistemic modality the author intends to follow in this work.
• Also, (1) and (2) are structurally different: (1) is a subject control construction while (s) involves the complement-taking verb juede that selects a clause as its complement. It is not clear to me whether the author takes the structural differences into consideration. Also, what distinguishes (2) from (1) is that there is an overt subject in the embedded clause selected by the verb juede. Let’s assume that the subject in the embedded clause plays a role in distinguishing the two types here. It is not clear how the author classifies the data from (3) to (9). What follows is a table to summarize the structural differences between the seven functional categories of we juede the author adopts from Chang (2016) in Section 2.4. It follows that T7 is structurally different from the others. I am wondering how this structural difference is interpreted along the line of the author’s reasoning that ‘T7 seems indicative that the children have become more conscious of differentiating self and others by this stage in development’ (p.12).
|
|
Subject control |
Embedded clause |
|
T1: Agreeing |
(3) |
|
|
T2: Disagreeing |
|
(4) [the embedded subject biyeluxing] |
|
T5: Commenting/Reasoning |
|
(5) [the embedded subject chuan bianfu] |
|
T4: Speculating |
|
(6) [the embedded subject yinliao] |
|
T5: Suggesting |
|
(7) [the embedded subject laoshi] |
|
T6: Concluding |
|
(8) [the embedded subject lengyin] |
|
T7: Afterthoughts |
|
(9) [if-clause] |
• If my understanding is correct, the author seems to focus on the type (2) rather than (1). If there is any particular reason to exclude (1) from the current study? The comparison between (1) and (2) can be of great importance along the lines of discussion in the current work.
• From line 45-54, the author provides a brief overview of previous studies on wo juede but the discussion does not suffice to show what the research gap is and the significance of these studies. What implications can these studies provide to recognize the research gap the author attempts to identify.
• It is suggested that the author should a brief overview of we juede or similar expressions investigated in previous studies from the perspective of language acquisition or discourse/pragmatics. Otherwise, this paper seems to be a work that reduplicates Chang’s (2006) work by putting itself in the context of Taiwan Mandarin-speaking children and adults.
• I am wondering whether there is any L1/L2 acquisition study on the seven functions of wo juede; that is, these functions are not specific to the expression wo juede but reflect cognitive functions. This line of discussion can support the view the author puts forward on p.12 ‘On closer inspection…’
• T6 is less the frequently used category by the child groups, as the use of this concluding function hinges on information integration in the immediate context. If there is any empirical acquisition study adding weight to this line of thinking, like the acquisition of textual/conversational cohesion/cohere or transitional expressions whose use is sensitive to discourse?
• On p.13: Can the use of a connective suoyi ‘so’ signal the processing time necessary for the children to integrate the conversational information in the context? If yes, this connective can be another discourse marker signaling processing load needed for information integration. Then, can this connective be analyzed on a par with wo juede?
• On p.13-14: As reported by the author, the children rarely put wo juede at the end of clause. As mentioned above, I am wondering whether this can be ascribed to structural difficulties; that is, the embedded clause encoding new information(focus) or old information (topic) is displaced to a sentence-initial position, which is a common operation in Mandarin syntax.
• The two construction patterns the author reports in Section 3.3 are thought-provoking. But the discussion in this section is not detailed. I suggest that the author could consider relating the section to discourse analysis/pragmatics to show that the use of wo juede is sensitive to turn-taking functions or utterance-initial/-final positions (i.e., foregrounding or backgrounding, if the author is not syntax-oriented), with respect to the two groups.
• On the other hand, it is suggested that the author could provide a crosslinguistc or cognitive perspective to show the seven functions are not genre-specific but a general cognitive development.
Minor comments:
Line 289: *nodoubt –
Line 351: * diversifiedpattern
Reviewer 2 Report
please check the attachment

Reviewer 3 Report
This study investigates the development pattern of the formulaic pattern “wo juede” in L1 Taiwan Mandarin by comparing the elicited spoken discourse produced by adults and primary school students from three different grades in argumentative and negotiative tasks. The results show that while older participants appeared to be able to use “wo juede” in more various contexts, all four groups tended to produce more instances of “wo juede” in the argumentative task and preferred to use this formulaic expression to achieve the purpose of commenting or reasoning.
Although this article lays out the research design and results in a fairly clear way and the findings are interesting, a number of issues need to be addressed before being considered for publication:
Abstract and Keywords
1. Since the author doesn’t mention Chang’s (2016) in the abstract, it is advised that the author refer to the function of commenting/reasoning as it is instead of T3, or refer to the function as T3 while mentioning Chang (2016) in the abstract.
2. It seems that Piaget’s theory of cognitive development plays quite an important role in this study, the author may consider including some terms related to Piaget’s theory in the abstract and in the keywords. (Taiwan Mandarin and discourse markers don’t seem to be that important in this article.)
Introduction
3. The author claims in the first sentence of the introduction that “[o]ne of the underexplored lexical and morphological issues from the perspective of discourse and grammar is that of fixed expressions or formulaic language.” If expressions such as “wo juede” are considered formulaic language in this study, then formulaic language is definitely one of the most well-studied topics in the field of discourse and grammar across languages (Hakulinen & Selting, 2005; Hsieh, 2018; Kärkkäinen, 2003, 2012; Tao, 2020; Wu & Biq, 2011, among many others). The author needs to rephrase that very first statement in order to be more accurate about the status quo of the literature.
4. The author may have to present a more substantial and organized literature review on topics including “wo juede”, acquisition of cognitive verbs in Mandarin or English, and development/acquisition patterns of formulaic language. The present version doesn’t seem sufficient.
5. Although the importance of formulaic patterns to the research of morphology has been alluded to several times throughout the article, the author doesn’t really identify how the research of formulaic language development in general or the current study in particular contributes to our understanding of Chinese morphology (development). S/he needs to specify the relevance of this study.
6. Although the author refers to “the first research question” in 3.1 (p. 8), the exact research questions can’t really be found in the text. The author needs to specify the research questions, probably in the introduction.
7. Similar to point 2 above, the author needs to mention Piaget’s theory in the introduction due to its importance to the current study.
Research Design
8. The author needs to explain why children in the concrete-operational stage were chosen to be the participants. How is the characteristic of this stage relevant to the aim of this research?
9. The author needs to elaborate on what “negotiate/negotiative/space to negotiate” means in this study. Did the participants really negotiate more in the negotiative genre?
10. If space allows, the author may consider providing examples of the prompts for both the argumentative and negotiative task. Table 3 seems to be an example of the negotiate task (the author needs to specify that as well).
11. Chang’ (2016) seems to put forth the most important framework for the current study. If that’s the case, the author needs to present a more in-depth review on that study (e.g., data, methodology, and main findings) and explain why Chang’s framework, among several others, is adopted.
Results and Discussion
12. The author needs to explain why “argumentative genres are of inter-subjectification, while negotiative genres are more of subjectification” (argumentative seems to be more subjective and negotiative sounds more intersubjective) and why (inter-)subjectification, instead of “(inter-)subjective,” is used, given that (inter-)subjectification seems to imply a process and (inter-)subjective is more about the quality. None of the above is self-explanatory.
13. The author makes several links between the pragmatic functions of “wo juede” and cognitive patterns of the participants. Are the links based on Chang’s (2016) writing or made primarily by the author? If the links are primarily made by the author of the current study, then on what grounds? Are they supported by any previous studies? The author may have to cite some studies to support the analysis.
14. Although the author does provide a definition for the term “construction tendencies” in the beginning of 3.3, it is still not clear which framework the author is adopting. Is it constructions as theorized in the model of Construction Grammar (Goldberg, 1995)? Based on the context and the definition, the author appears to lean toward the model of Conversation Analysis or Interactional Linguistics. However, this doesn’t seem to be how CA or IL researchers use the term “construction”. Hence, the author may need to cite sources or elaborate on the meaning of the term.
15. The author should also mention constructional tendencies in one of the research questions or the research design if that is one of the main parts of the study (as it constitutes the main topic of an entire sub-section). And if the author is taking a CA/IL approach, this should also be revealed in the introduction or research design section.
16. The reader of this article may not be familiar with the idea of TCU mentioned 3.3 on page 13. The author may need to briefly explain what it means either in the main text or in a footnote (as it may not (always) be equal to a clause). Or, at least one reference (e.g., Sacks et al., 1974) should be provided for the definition of this term.
Discussion and Conclusion
17. The author seems to use “discussion” twice in the section headings, which is not very conventional, to say the least.
18. The author refers to one of the oral production tasks in the second line of the discussion and conclusion section as descriptive genres. Does the author mean negotiative genres? The author doesn’t seem to refer to the task as a descriptive genre. Are these two terms interchangeable?
19. In the discussion and conclusion section, the author should elaborate more on the contributions of this study. What are the implications of the findings for the research of formulaic language development/acquisition and of Chinese morphology?
20. If space allows, the author should also identify some of the limitations of the current of the current study, e.g., the age groups involved in the study. We don’t really know when L1 Taiwan Mandarin speakers start to use wo juede as a fixed/formulaic expression in their discourse and when L1 Taiwan Mandarin speakers acquire the more advanced function(s) of wo juede.
21. There are also a few typographical errors. For example,
“argumentativegenres” (p. 8)
“nodoubt” (p. 8)
“diversifiedpattern” (p. 10)
“smallerscale” (p. 13)
My comments end here.
References:
Hakulinen, A., & Selting, M. (Eds.). (2005). Syntax and lexis in conversation: Studies on the use of linguistic resources in talk-in-interaction. Amsterdam: John Benjamins.
Hsieh, C. Y. C. (2018). From turn-taking to stance-taking: Wenti-shi ‘(the) thing is’ as a projector construction and an epistemic marker in Mandarin conversation. Journal of Pragmatics, 127, 107-124.
Goldberg, A. (1995). Constructions. A Construction Grammar Approach to Argument Structure. Chicago: University Chicago Press.
Kärkkäinen, E. (2003). Epistemic stance in English conversation. A description of its interactional functions, with a focus on I think. Amsterdam: John Benjamins.
Kärkkäinen, E. (2012). I thought it was very interesting. Conversational formats for taking a stance. Journal of Pragmatics, 44(15), 2194-2210.
Sacks, H., Schegloff, E. A., & Jefferson, G. (1974). A simplest systematics for the organization of turn taking for conversation. Language, 50(4), 696-735.
Tao, H. (2020). Formulaicity without expressed multiword units. In Ritva Laury and Tsuyoshi Ono, eds., Fixed Expressions: Building language structure and social action, 71-98. Amsterdam: John Benjamins.
Wu, A. Y. R., & Biq, Y. O. (2011). Lexicalization of intensifiers: Two X-shi constructions in spoken Mandarin. Chinese Language and Discourse, 2(2), 168-197.
Reviewer 4 Report
Since there have already been several studies of ‘wo juede’ in the literature, a stronger justification for working on the same topic would be welcome. It’s true that the author(s) have justified this by saying that their focus is on child language acquisition in Taiwan Mandarin. Still, it would be good to see a more sustained argument for doing another major study of ‘wo juede’ that highlights the ‘argumentative’ vs ‘negotiative’ distinction. How important or basic is this distinction in the context of the full range of uses of ‘wo juede’? And what differences, if any, are there between Taiwan Mandarin and other varieties of Mandarin in terms of its use of ‘wo juede’?
Regarding the seven functions of ‘wo juede’, as the authors acknowledge in the paper, these functions are not always readily distinguishable without due consideration of the context. I wonder if the coders were always able to agree on the coding, given the difficulty of the task? Were there instances where even after a third coder was brought in it was still not clear which function was the ‘correct one’ in a given context?
Another question about the function is how they are defined in the first place. Specifically, there would seem to be a question of their comparability, i.e., are they really comparable as alternative actions on the same plain/footing? Thus, ‘comment/reasoning’ and ‘agreement/disagreement’ would seem to be much broader categories than ‘afterthoughts’. If so would their frequencies of occurrence be intrinsically different anyway?
Another issue is to do with the authors’ claim that based on the findings one can conclude that children are less able or competent to perform “deductive logic”. However, as the authors themselves observe, children often use ‘suoyi’ to introduce logical conclusions. If so then why is the less frequent use a sign of inferior ability to do “deductive logic”?
Finally, the argument that this paper makes a good contribution to a special issue on Chinese morphology may need to be further substantiated, as the relevance of this study of morphology is not immediately self-evident.
Reviewer 5 Report
This article examines the production of the expression ‘wo juede’ by primary school pupils whose native language is Taiwan Mandarin. Some Grade 2, 4 and 6 pupils and college students were recruited in this study to engage in discussions on several selected topics in small groups. Their use of the expression ‘wo juede’ in the discussions were analyzed in terms of frequency, functional types, and cross-group comparisons. The author(s) found that all informants use the expression ‘wo juede’ more frequently in discussions with argumentative topics than with negotiative ones, and they are more inclined to use this expression for commenting/reasoning than the other six pragmatic functions. The patterns of using this expression among children of different age groups are also observed: the use of ‘wo juede’ for speculating and suggesting are first detected at the intermediate grade (G2), and the use of it for afterthoughts are first detected at the upper grade (G3), but the use of it for concluding is not found among pupils in this study.
This paper reports a well-organized experiment examining the use of a most commonly used expression ‘wo juede’ in spoken Mandarin. However, I am very concerned about the conceptual framework adopted in this study, the analysis of the collected data, and thus the extent to which the experimental findings can offer to the insights to the research topic.
1. The use of terminology and the conceptual framework used in this article is very weak. In this study, the expression ’wo juede’ is referred as a fixed expression. According to Baker (1992: p.63), ”idioms and fixed expressions are at the extreme end of the scale from collocations in one or both of these areas: flexibility of patterning and transparency of meaning. They are frozen patterns of languages which allow little or no variation in form and, in the case of idioms, often carry meanings which cannot be deduced from their individual components”. According to Baker’s definition of idioms and fixed expressions, ’wo juede’ cannot be a fixed expression.
The author(s) adopted the seven functional categories from Chang (2016) as the conceptual framework for analyzing the functions of 'wo juede', however, Chang (2016) is an unpublished MA thesis, which was not subjected to peer review. Considering that the content cited from Chang (2016) forms a significant part of the current work that they are trying to publish, it is inappropriate to do so.
Baker, M. (1992). In other words: A coursebook on translation. London and New York: Routledge.
2. The author(s) applied statistical tests to the data, but given the limited number of cases, many of the results from the statistical analysis are meaningless. For example, the chi-square test was used to examine the within-group differences in Table 7. For Group 1, T3 (6 cases) is compared with T1/T2 (1 case each), or T3 (6 cases) is compared with T4-T7 (0 case), it is pointless to use the chi-square as it requires the minimum of 5 cases for each group under comparison. Many of the statistical results listed in Table 5 are also meaningless.
3. The scholarly significance and contribution of this work is over exaggerated. In section 4, paragraph 2, the author(s) stated the following: ”This study has contributed to the study of Chinese morphology by raising awareness about the acquisition and development patterns in the area of fixed expressions or formulaic language”(p.14). I am afraid I cannot agree with it at all.
Therefore, I cannot approve the manuscript in this form.
Round 2
Reviewer 1 Report
The current draft is greatly improved and the author has replied to most of my comments in the current draft, (maybe not all of my comments because the version the author uploaded contains many crosslines and track changes, making it hard to read the draft with ease). What is more, no reply report is provided. Thus, it is difficult to see whether the author did reply to every of my comments.
Overall, I recommend that this manuscript CANNOT be published in its current form unless it is proofread by a native speaker of English to improve his/her English writing in this draft and the author needs to make sure that the manuscript is error-free.
****I spotted some errors/inconsistencies for the author’s reference.
In the abstract: I think that we do not say ‘native Mandarin Chinese’, right?
On p.18: the concluding and after-thought groups(?) ïƒ the concluding and after-thought category?
groups or categories?
Here we find that among the seven categories, two of them, the concluding and after-thought groups,
On p.21: a grammatical error— lack of a conjunction
……One is that this study has a modest sample size, more intensive studies will be needed to uphold any legitimate claims. ….
On p.2: one groups ïƒ one grop
……Aiming to fill the research gap in L1 acquisition of Taiwan Mandarin, we examine the use of WJ in three groups of children (Grades 2, 4, and 6) and one groups of adults for comparison…
Reviewer 2 Report
The revised version is clear in organization and argumentation.
Author Response
Thank you.
Reviewer 3 Report
The manuscript has undergone substantial revision according to the reviewers' comments.
Author Response
Thank you.
Reviewer 5 Report
The language has been improved significantly in the revised version, more previous literature has been added, especially concerning the discussion of WJ as a fixed expression.
However, only literature concerning WJ were mentioned, the author(s) did not mention any previous studies regarding the acquisition of Chinese words or expressions. The author(s) claim ‘Aiming to fill the research gap in L1 acquisition of Taiwan Mandarin…’(p. 3), which I am highly doubtful about. Do they mean there is no previous studies about L1 acquisition of Taiwan Mandarin at all? Even if that is the case, they can quote similar literature in the area of L1 acquisition of Mandarin Chinese, since this is the main theme of this study.
Regarding the issue raised regarding the use of statistical tests in the first round of review, the author(s) provided response, which I unfortunately do not agree.
The scholarly significance and contribution of this work is over exaggerated in the abstract and the conclusion section.
